# Fermented Rapeseed and Soybean Alone and in Combination with Macro Algae Inhibit Human and Pig Pathogenic Bacteria In Vitro

**DOI:** 10.3390/microorganisms12050891

**Published:** 2024-04-29

**Authors:** Frederik Beck, Ninfa Rangel Pedersen, Dennis Sandris Nielsen

**Affiliations:** 1Fermentationexperts, Vorbassevej 12, 6622 Baekke, Denmark; frederik.beck@food.ku.dk (F.B.); nrp@fexp.eu (N.R.P.); 2Department of Food Science, University of Copenhagen, Rolighedsvej 26, 1959 Frederiksberg C, Denmark

**Keywords:** antimicrobials, fermentation, pathogens, feed ingredients, pig farming, gut health

## Abstract

Higher plants produce secondary metabolites expressing antimicrobial effects as a defense mechanism against opportunistic microorganisms living in close proximity with the plant. Fermentation leads to bioconversion of plant substrates to these bioactive compounds and their subsequent release via breakdown of plant cell walls. Fermented feed products have recently started to become implemented in the pig industry to reduce overall disease pressure and have been found to reduce events such as post-weaning diarrhea. In this study, we investigate the antimicrobial potential of fermented soybean- and rapeseed-based pig feed supplements with and without added seaweed. The antimicrobial effect was tested in a plate well diffusion assay against a range of known human and livestock pathogenic bacteria. Further, we investigate the metabolite profiles based on liquid-chromatography mass-spectrometry (LC-MS) analysis of the fermented products in comparison to their unfermented constituents. We observed a pronounced release of potential antimicrobial secondary metabolites such as benzoic acids when the plant material was fermented, and a significantly increased antimicrobial effect compared to the unfermented controls against several pathogenic bacteria, especially *Salmonella enterica* Typhimurium, *Listeria monocytogenes*, *Yersinia enterocolitica*, and a strain of atopic dermatitis causing *Staphylococcus aureus* CC1. In conclusion, fermentation significantly enhances the antimicrobial properties of rapeseed, soybean, and seaweed, offering a promising alternative to zinc oxide for controlling pathogens in piglet feed. This effect is attributed to the release of bioactive metabolites effective against pig production-relevant bacteria.

## 1. Introduction

Upon weaning, pigs often display increased susceptibility to the development of diarrhea. This condition is attributed to stress induced by changes in the environment, social regrouping, and the transition from maternal milk to solid feed [1]. Additionally, a reduction in nutrient absorption is observed as the pigs have to adapt to solid feed. Historically, post-weaning diarrhea was controlled by incorporating antibiotics as growth promoters in pig feed. However, due to escalating instances of antibiotic resistance, the European Commission prohibited such preventative measures in 2006 [2]. Subsequently, pharmacological zinc oxide (ZnO) was implemented as an alternative compound for reducing post-weaning diarrhea. However, due to increasing concerns of environmental risks [3], and studies linking ZnO with development of antibiotic resistance [4], feed supplementation of pharmacological ZnO above 150 ppm became prohibited in the European Union as of June 2022 [5]. This has necessitated the urgency for development of new strategies to preserve health in weaner piglets.

A major challenge associated with post-weaning diarrhea is the significant reduction in nutrient acquisition that adversely impacts growth rates [6]. A strategy to circumvent the lowered nutrient uptake is to implement fermented ingredients into the feed. Typically, these fermentation processes are conducted using lactic acid bacteria [7,8,9]. This leads to the breakdown of structural components of the feed and release of essential nutrients from the plant material [10], increasing the bioavailability of these nutrients in the pig gut [11,12]. Additionally, fermentation leads to release of bioactive plant and microbial secondary metabolites with antimicrobial properties against pathogenic bacteria [13]. Notably, plant metabolites such as polyphenols and other organic compounds such as benzoic acids exhibit these antimicrobial effects [14,15,16,17].

Research indicates that fermented rapeseed and soybean meal significantly decreases diarrhea incidence in weaned piglets when added to the feed [7]. A study by Xie et al. (2017) demonstrated that 21 days of feeding fermented soybean meal (FSBM) led to a decrease from 13.7% (control group) to 8.8% (FSBM fed) in diarrhea incidence. Interestingly, the relative abundance of Prevotella, lactobacilli, and butyrate-producing bacteria in fecal samples also increased in the FSBM-fed pigs [18]. The short-chain fatty acid (SCFA) butyrate functions as an energy source for the gastrointestinal epithelial cells and promotes maintenance and renewal of the epithelium [19,20]. Xie et al. (2017) found that the intestinal epithelium benefitted from supplementation of fermented soybean meal (15%), evidenced by increased length of the epithelial villi and depth of the villi crypts compared to weaned piglets fed a regular non-fermented corn-soybean diet [18]. The improved health status observed in the weaned piglets fed the fermented soybean meal is likely caused by suppression of opportunistic pathogenic bacteria known for causing infections in the delicate gut tissue. This happens due to competitive exclusion by commensal bacteria, and their gut-health-promoting properties, and increased gut health due to increased production of SCFA such as butyrate, from breakdown of indigestible carbohydrates [21,22,23]. Furthermore, the elongation of villi increases the surface area available for nutrient absorption, thereby enhancing nutrient uptake efficiency, which is vital for optimal growth and health [24,25]. Additionally, the presence of shallower crypts suggests a diminished requirement for cellular turnover and regeneration, indicative of a more stable and healthy gut lining [26].

Macro algae contain various compounds with antimicrobial activity, including polypeptides, fatty acids, benzoic acids, carotenoids, and polyphenols [27,28,29]. Hui et al. (2021) demonstrated how supplementation of fermented feed based on rapeseed meal in combination with the brown algae *Ascophyllum nodosum* and *Saccharina latissima*, altered the microbiome composition of weaner piglets and improved gut health [9]. Similarly, studies by Satessa et al. (2020) and Hui et al. (2021) also revealed gut health-promoting properties of fermented rapeseed meal alone and in combination with the seaweed *Ascophyllum nodosum*, evidenced by increased gut villi length and villi crypts depths, which are markers of a healthy gastrointestinal epithelium [8,9].

Higher plants synthesize secondary metabolites as a defense mechanism against pathogens [30]. These metabolites can be harnessed as growth-inhibiting agents against specific pathogenic microorganisms in various applications, including animal feed [15,31,32]. Additionally, fermentation facilitates an increased release of bioavailable compounds of the plant material [13,33,34]. However, it remains to be investigated how fermentation may influence the antimicrobial properties of rapeseed and soybean meal. This study aims to elucidate the antimicrobial effects of fermented rapeseed meal and soybean meal both individually and in combination with macroalgae, against a range of different pathogenic bacteria. We hypothesize that fermented plant materials such as soybean and rapeseed meal alone and in combination with fermented seaweed exhibit an increased antibacterial effect against pathogenic bacteria, compared to their unfermented products.

## 2. Materials and Methods

### 2.1. Preparation of Extracts of Fermented Products for Investigation of Antimicrobial Activity

The fermented products and the unfermented raw materials, investigated for antimicrobial activity, were supplied by European Protein (Baekke, Denmark). The fermentation process detailed by Satessa et al. (2020) [7] involved a bacterial triad consisting of *Pediococcus acidilactici* (DSM 16243), *Pediococcus pentosaceus* (DSM 12834), and *Lactiplantibacillus plantarum* (DSM 12837). This was conducted in a solid-state anaerobic process and subsequently dried [7]. All products and unfermented raw materials (Table 1) were dissolved (1:5) in three distinct organic solvents of different polarity: dimethyl sulfoxide (DMSO) (polar), ethyl acetate (semi-polar), and n-hexane (non-polar) [35]. This dissolution was maintained for five days at room temperature, with the flasks being gently shaken 8–10 times daily. The supernatant was filtered into new sterile flasks through 4-layer 28-thread filter cloth (Valdemar Larsens Eftf. A/S). The extracts were stored at 5 °C and tested within two days of extraction.

### 2.2. LC-MS Analysis

The metabolites of the investigated products were qualitatively mapped using liquid chromatography mass spectrometry (LC-MS), conducted externally at MS-Omics Aps, Vedbæk, Denmark. The samples were weighed in cryotubes and shipped to MS-Omics, and all extractions and preparations of the samples for the LC-MS analysis were performed in the semi-polar solvent ethyl acetate. The products seen in Table 1 were analyzed together with the controls consisting of unfermented rapeseed and soybean meal and the two brown algae *Ascophyllum nodosum* and *Saccharina latissima*, which were also co-fermented in some of the tested products. The results of the analysis are shown as calculated log-2-fold changes of each metabolite, across the different products tested.

### 2.3. Bacterial Strains and Propagation Conditions

The antimicrobial potential of the fermented products was tested against representatives of five different bacterial pathogens of concern in the pig industry and human health aspects: *Salmonella enterica* Typhimurium, *Listeria monocytogenes*, *Escherichia coli*, *Yersinia enterocolitica*, and *Staphylococcus aureus* (Table 2). All bacterial strains were sourced from the in-house culture collection at the University of Copenhagen, Department of Food Science, which are stored in stock-cultures at −80 °C. For propagation, the bacteria were streaked on brain/heart infusion (BHI) agar (HiMedia Laboratories GmbH, M210-500G) plates and incubated at 37 °C for 24 h. Subsequently, one bacterial colony was inoculated into 10 mL BHI broth and incubated for 24 h at 37 °C under shaking conditions (250 rpm).

### 2.4. Plate Well Diffusion Assay

The prepared extracts were assessed for their antimicrobial potential using a plate well diffusion assay [36]. A volume of 450 µL of each bacterial suspension propagated as described previously was inoculated into 500 mL of cooled liquid agar (48 °C) and then poured into 90 mm Petri dishes and allowed to solidify for 30 min. Wells of 6 mm in diameter were punched out in the solidified agar. These wells were filled with 200 µL of the extract, which was allowed to diffuse into the agar under incubation at room temperature for two hours. The solvents (ethyl acetate, n-hexane, and DMSO) functioned as negative controls. After incubation at 37 °C for 24 h, the growth inhibition zones were measured (mm) using a transparent ruler. The measured clearing zones of the negative controls were subtracted from the clearing zones of the extracts. All tests were conducted with five biological replicates, and the data are presented as a calculated mean ± standard deviation.

### 2.5. Thermal Stability and Resistance to Proteolytic and Lipolytic Enzymes of Antimicrobial Compounds

To assess the thermal stability of the potential antimicrobial compounds, four aliquots of each filtered solvent were heat -treated at 40, 60, 80, and 100 °C for 30 min. Lipase and proteolytic enzyme sensitivity were tested by the addition of lipase (Sigma Aldrich, St. Louis, MO, USA, CAS no.:9001-62-1), proteinase K (Sigma Aldrich, CAS no.: 39450-01-6), or trypsin (Sigma Aldrich, CAS no.: 9002-07-7) to the extracts, all in a concentration of 1 mg/mL and incubated for 30 min at 50 °C (enzyme activation temperature). The treated extracts were then reassessed for their antimicrobial effect in an agar well diffusion assay similar to the setup described above against two strains of *L. monocytogenes* and *S.* Typhimurium, respectively, representing Gram-positive and Gram-negative pathogenic bacteria.

### 2.6. Statistics and Data Visualization

Extensive data visualization and analysis were conducted utilizing tools provided by the R programming language. Plots and visual representations were generated using several R-packages, including ggplot2 [37], tidyverse [38], cowplot [39], and ggpubr [40]. Statistically significant differences between datasets were determined using the non-parametric Wilcoxon signed-rank test. This test, renowned for its robustness in handling non-normal data distributions, was executed using the built-in ‘Stat_compare_means’ function in R, a tool that provides an efficient and accurate means for statistical comparison of means in various groups. For the visualization of complex datasets, heatmaps were generated using the ‘Autoplot’ function in R. This function excels at displaying multi-dimensional data, thereby enabling a clearer understanding of the underlying patterns and relationships within the dataset.

## 3. Results

### 3.1. Extracts of Fermented Plant Material Contain Higher Amounts of Secondary Metabolites, Relative to Unfermented Control Products

Metabolomics analysis of the fermented plant materials assessed by liquid chromatography-mass spectrometry (LC-MS), revealed a significant increase in the abundance of metabolites across all analyzed structural classes in the fermented variants (Figure 1A–E). The fermented products exhibited a higher relative abundance of metabolites in comparison to their non-fermented controls (Figure 1). Specifically, both fermented rapeseed meal (FRSM) and fermented soybean meal (FSBM) demonstrated an increased relative abundance of the majority of metabolites assessed relative to the unfermented controls, rapeseed meal (RSM) and soybean meal (SBM), respectively. Among the non-fermented products, RSM displayed a greater abundance of most metabolites compared to SBM. Additionally, the combination of fermented rapeseed meal with seaweed (FRSMS) and fermented soybean meal with seaweed (FSBMS) also exhibited a higher relative abundance of metabolites than the unfermented components comprising these mixtures.

Overall, benzoic acids were observed in higher abundance in the fermented products compared to the unfermented controls (Figure 1A), with the two fermented seaweeds *A. nodosum* and *S. latissima* were found to have the highest relative amount of 2,3,4-trihydroxybenzoic acid compared to the rest of the analyzed samples. A similar trend was further observed in the analysis of the fatty acid profile (Figure 1B), again with *S. latissima* showing greater amounts of the fatty acids (13/9(S)-HODE (C18:2), traumatic acid (C18:1), and lauric acid (C12:0)) compared to the rest of the samples. In the analysis of carbohydrates (Figure 1C), we observed that in rapeseed meal (RSM), several hexose dimers and trimers, as well as deoxyhexose, were relatively more abundant when compared to the fermented products and SBM. The fermented seaweed *A. nodosum* and *S. latissima* exhibited greater relative abundance of deoxyhexose compared to all other samples. Regarding indolyl carboxylic acids (Figure 1D), we observed a greater relative abundance of imidazole acetic acid in the fermented products based on rapeseed meal (FRSM and FRSMS) relative to the other substrates and products. When comparing FSBM and FSBMS, we found that indole-3-glyoxylic acid was more abundant in FSBMS. The same compound was also found in relatively high amounts in the unfermented algae *S. latissima*, one of the components of FSBMS.

The pattern of elevated metabolite abundance in *S. latissima* was observed in the amino acid profile analysis (Figure 1E). Notably, elevated abundance of proline, a non-essential amino acid with key roles in protein structure/function and maintenance of cellular redox homeostasis [41], and essential amino acids such as phenylalanine, alanine, leucine valine, and isoleucine were observed in unfermented *S. latissima.*

### 3.2. Fermented Soybean- and Rapeseed-Based Supplements Exhibit Greater Inhibition of Microbial Pathogens Compared to Unfermented Controls

Agar well diffusion assays were conducted to determine the antimicrobial effects of fermented products and the unfermented control products against *S. aureus, L. monocytogenes, S.* Typhimurium, *E. coli*, and *Y. enterocolitica*. Among the three extraction solvents used, extractions with ethyl acetate resulted in the highest pathogen inhibition with an average of 0.8 mm (negative controls subtracted). In comparison, extractions with dimethyl sulfoxide (DMSO) exhibited no inhibition. Similarly, products extracted with n-hexane, as well as treatment with n-hexane alone (control), did not inhibit any of the five species. Consequently, subsequent experiments were focused solely on extracts obtained using ethyl acetate.

The fermented products exhibited significantly greater inhibition towards all three strains of *S.* Typhimurium (Figure 2D–F) compared to the unfermented control products with several-fold larger clearing zones when comparing FRSM and its unfermented counterpart against *S. enterica* 1575. A similar pattern was observed for *L. monocytogenes* (Figure 2A–C) as the fermented products were shown to have a significantly higher level of inhibition compared to the unfermented. The three strains of *Yersinia enterocolitica* exhibited a tendency for increased antimicrobial potential in the fermented products compared to the unfermented controls (Figure 2G–I). Varying results were obtained for *Staphylococcus aureus*, where the fermented products showed significantly greater inhibition against the indicator strain *S. aureus* CC1, compared with the unfermented controls (Figure 2K). However, no significant effects were observed, when testing against *S. aureus 1077* (Figure 2J). For the four strains of *E. coli*, clearing zones smaller than 1 mm were observed, after the negative solvent control clearing were subtracted. Additionally, the reproducibility of results for *E. coli* was relatively low, and no consistent pattern of inhibition was evident between the fermented and unfermented products. Generally, Gram-positive bacteria were more susceptible to the antimicrobial effects of the fermented products than Gram-negative ones. The fermented rapeseed and soybean products demonstrated similar antimicrobial potentials against the tested pathogens. The inclusion of seaweed in the fermented products did not significantly enhance their antimicrobial properties.

### 3.3. Heat Treatment of Extracted Metabolites Lowers the Antimicrobial Potential

To further investigate the physiochemical properties of the compounds that potentially inhibit growth of the tested pathogens, we subjected the extracts to proteolytic and lipolytic enzymes as well as elevated temperatures for heat inactivation of the putative microbial inhibitory compounds. Treatment with proteolytic or lipolytic enzymes generally did not decrease activity in any of the tested extracts, other than treatment with trypsin, which eradicated the growth inhibition seen in *L. monocytogenes*. However, these results are inconclusive as the untreated samples only led to inhibition zones of 0.5 mm, which is not considered indicative of significant inhibition in this assay (Appendix A). Furthermore, the heat inactivation assay conducted on the four products showed a clear decrease in the antimicrobial potential (Figure 3). When testing the heat-inactivated extracts on the *L. monocytogenes* strain, the products based on rapeseed did not show any clear patterns of decrease in inhibition, as the untreated control resembled all of the extracts across the temperature gradient. We did observe an increased inhibition of the extracts from the soybean-based products *on L. monocytogenes* that did not undergo heat treatment compared to extracts from the heat gradient that showed decreasing antimicrobial effects as the temperature increased in the analysis. Focusing on the heat stability of the antimicrobial compounds against *S.* Typhimurium, we did overall find a decrease in inhibition over the temperature gradient, and a noticeably higher inhibition was observed in the untreated samples compared to the heat-treated samples, especially 100 °C for 30 min which led to loss of antimicrobial activity (Figure 3).

## 4. Discussion

This study was designed to investigate the antimicrobial potential of fermented pig feed supplements, based on soybean and rapeseed, with and without seaweed. By comparing the antimicrobial effect of fermented and unfermented products, we also sought to investigate the exact contribution of the fermentation process.

The distinction between fermented and unfermented plant materials is evident from the metabolite profiles analyzed across various fermented products in comparison to their unfermented raw material counterparts. The relative concentration of various metabolites in different metabolite classes (amino acids, benzoic acids, carbohydrates, fatty acids, and indolyl carboxylic acids) were markedly higher in the fermented products relative to the unfermented controls. Several of the compounds, especially the benzoic acids, exerted antimicrobial activity. Of the benzoic acids, we know from literature that gallic acid exhibits antimicrobial effect against *S. aureus* [40]; that 3-phenyllactic acid inhibits growth of species such as *Staphylococcus aureus*, *Klebsiella oxytoca,* and *S.* Typhimurium [41]; and that anthranillic acid exhibits the same effect against *S. aureus, E.coli,* and *S. enterica* Typhimurium [42]. The same result for benzoic acid is what we have observed to increase in the fermented products relative to the controls. This suggests a correlation between the increased amount of antimicrobial secondary metabolites observed in the LC-MS assay and the antimicrobial effect that we observed in the agar well diffusion assay. The inclusion of macro algae *A. nodosum* and *S. latissima* in the feed supplement formulations increased levels of compounds like indole-3-glycoxylic acid. The same tendency was seen in the known antimicrobial compound 3,4-dihydroxybenzoic acid, in which the *S. latissimi*-containing product FSBMS contained higher amounts of this metabolite relative to FSBM, which has the same composition except for the lack of supplemented macro algae. At the single metabolite level, the addition of macro algae to the fermented feed supplements may influence the nutritional value, as metabolites such as amino acids, carbohydrates, and fatty acids, beneficial for the host metabolism, are increased in the fermented products.

Based on the results obtained in the antimicrobial plate assay, it is evident that soybean and rapeseed, both separate and in combination with macro algae, exhibit antimicrobial properties. These findings align with previous studies, showing that plant secondary metabolites have antimicrobial effects against a range of pathogenic Gram-negative and -positive bacteria [14,43]. This antimicrobial effect is notably enhanced when the products are fermented, as the fermented products have significantly increased antimicrobial effects compared to the unfermented controls. The most pronounced effect of fermentation is observed in the case of *Salmonella enterica Typhimurium.* Here, we observed a minimal effect of the unfermented controls, but significant inhibition when introducing extracts of the fermented products. This enhanced effect is possibly derived from the breakdown of cell wall constituents of the plant material and a release of intracellular secondary metabolites as well as production of new metabolites with antimicrobial effects [7,10,14,15,44,45]. We observed no significant differences between the fermented products regarding antimicrobial efficiency, indicating that for the antimicrobial effect, the fermentation process might be more important than the exact raw materials. This can be due to the nature of the LAB used in the fermentation process, which are known for producing bacteriocins and bacteriocin-like substances that inhibit growth of several pathogens [46,47]. In the study by van Heel et al. (2011), it was observed how isolated LAB strains from different food products had antimicrobial effects against several pathogenic bacteria, including the species *Listeria monocytogenes* and *S. aureus* [48]. This indicates that the antibacterial also can occur based on the presence of extracted bacteriocins from the fermentative bacteria used in the products [49]. Further research is needed to fully clarify if that is likewise the tendency in our study. Isolates of the bacteria used for fermenting the products tested in this study could be tested against the bacteria that we were investigating in the same manner as it was conducted by van Heel et al. (2011).

## 5. Conclusions

This study elucidates the enhanced antimicrobial capabilities of fermented rapeseed, soybean, and seaweed, and underlines their potential as natural inhibitors against pathogens critical to pig production. The fermentation process notably augments the antimicrobial efficacy of these plant materials compared to their unfermented counterparts, likely due to the generation and release of bioactive antimicrobial metabolites. These findings propose fermented plant materials as viable alternatives to pharmacological agents like zinc oxide, which faces regulatory restrictions in animal feeds. Further research should explore the specific metabolites responsible for the observed antimicrobial effects and their mechanisms of action at the molecular level. Investigating their impact on overall animal health and productivity, as well as the environmental implications of their widespread use, will be crucial. Additionally, extending this research to other livestock species could broaden the applicability of fermented feed additives, potentially revolutionizing animal agriculture and contributing to the mitigation of antibiotic resistance.

## Figures and Tables

**Figure 1 microorganisms-12-00891-f001:**
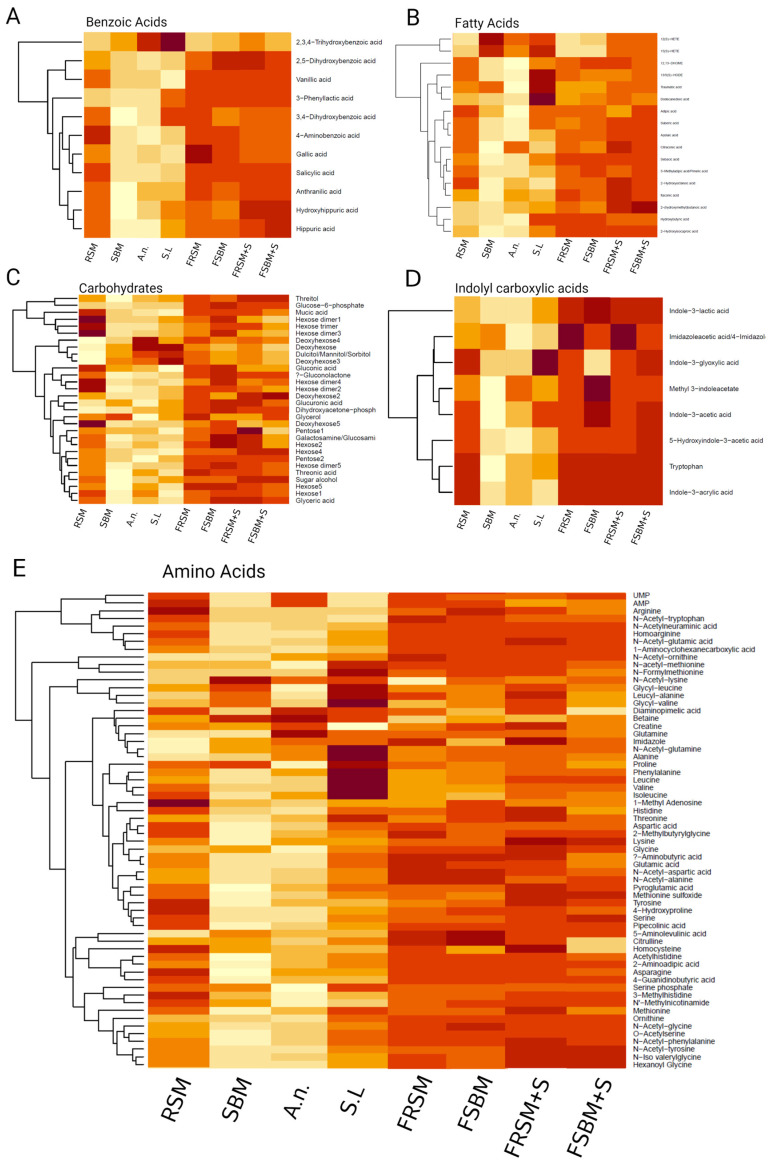
Variance of metabolite abundances across samples—Heat maps defining the relative abundance of major groups of metabolites analyzed in the LC-MS assay. In all groupings (benzoic acids (**A**), fatty acids (**B**), carbohydrates (**C**), indolyl carboxylic acids (**D**), and amino acids (**E**)), the different fermented products (FRSM = fermented rapeseed meal, FSBM = fermented soybean meal, +s = addition of seaweed) was compared with each other and the unfermented controls (RSM = unfermented rapeseed meal, SBM = unfermented soybean meal, A.N. = unfermented *A. nodosum*, S.L. = unfermented *S. latissima*.

**Figure 2 microorganisms-12-00891-f002:**
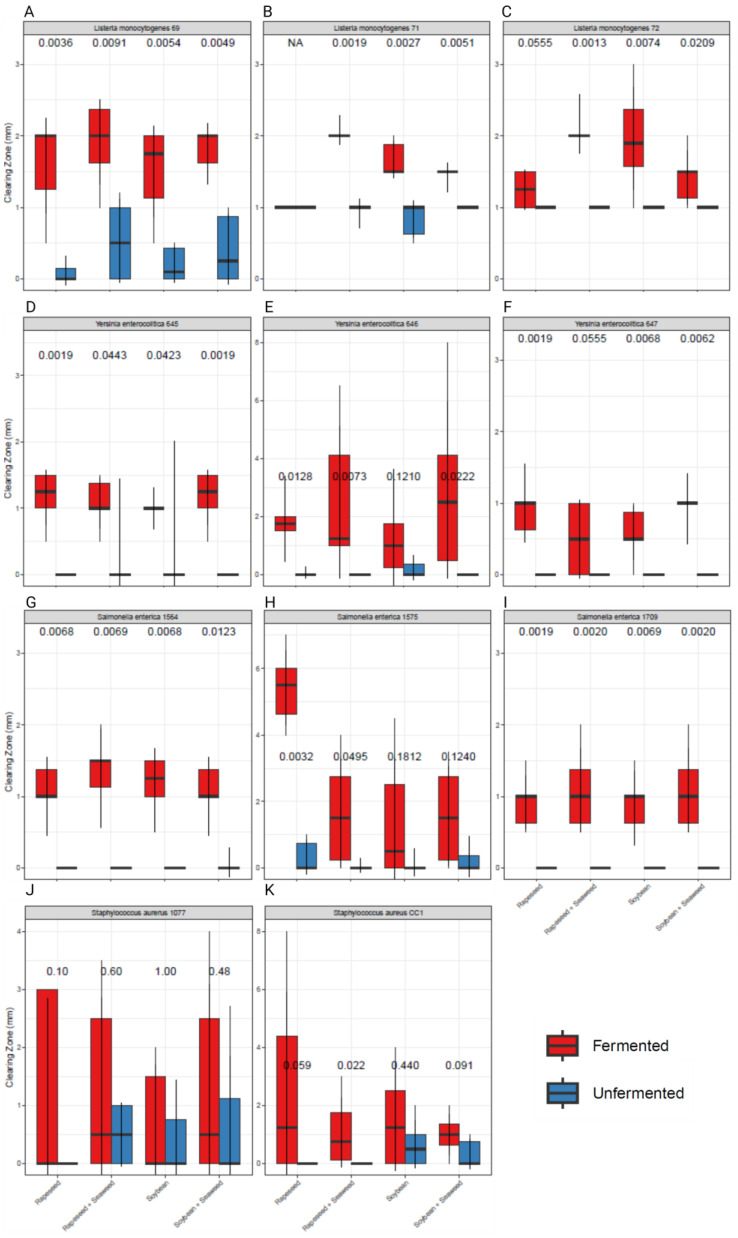
Side-by-side comparison dot plot of clearing zones observed in the agar well diffusion assays, in which extracts from fermented pig feed additives containing rapeseed and soybean meal, with and without seaweed and unfermented control products. The extracts are tested against the pathogenic bacteria *Listeria monocytogenes* (**A**–**C**), *Salmonella enterica Typhimurium* (**D**–**F**), *Yersinia enterocolitica* (**G**–**I**), and *Staphylococcus aureus* (**J**,**K**). Clearing zones were measured in mm from the edge of the well to the edge of the visible zone of complete bacterial growth inhibition. Error bars represent the standard deviation of the measured clearing zones in each product. Statistical differences were determined by non-parametric Wilcoxon signed rank tests.

**Figure 3 microorganisms-12-00891-f003:**
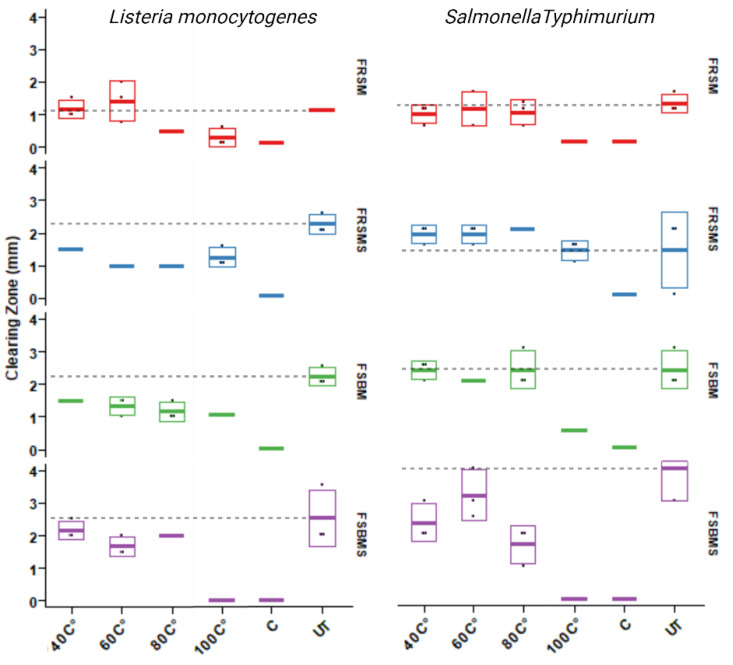
Heat inactivation of potential antimicrobial compounds—extracts of fermented products containing rapeseed (FRSM) and soybean meal (FSBM), alone and in combination with seaweed (FSBMS/FRSMS). Each of the extracts was heated at 40, 60, 80, and 100 °C and tested for its antimicrobial effects in a plate well diffusion assay against *Listeria monocytogenes* and *Salmonella* Typhimurium. Results of the heat treatment were compared to an untreated control (UT) and the extraction solvent ethyl acetate (C). Error bars visualize the calculated standard deviation.

**Table 1 microorganisms-12-00891-t001:** Overview of the different products being tested for their antimicrobial effects against known pathogenic bacteria. The products are separated into two categories based on the primary component (soybean or rapeseed meal). FSBM and FRSM are 100% fermented soybean and rapeseed meal, respectively. Both are also produced in variants with the addition of 12% seaweed in the fermentation (FSBMS/FRSMS).

Raw Material	Product	Content	Commercial Name
Soybean	FSBM	Fermented soybean meal	EP200
Soybean	FSBM/S	Fermented soybean meal + 12% seaweed (6% *S. latissima* + 6% *A. nodosum*)	EP2299
Soybean	SBM	Unfermented soybean meal	
Rapeseed	FRSM	Fermented rapeseed meal	EP100
Rapeseed	FRSMS	Fermented rapeseed meal + 12% seaweed (6% *S. latissima* + 6% *A. nodosum*)	EP1199
Rapeseed	RSM	Unfermented rapeseed meal	
Algae	AN	*Ascophyllum nodosum*	
Algae	SL	*Saccharina latissima*	

**Table 2 microorganisms-12-00891-t002:** Bacterial pathogens used for assessing the antimicrobial activity of fermented feed products.

Strain Number	Origin	Genus	Species	Subtype	Note
1575	Human	*Salmonella*	*enterica*	Typhimurium	Feces isolate from diseased individual
1564	Pig	*Salmonella*	*enterica*	Typhimurium	Feces isolate from diseased individual
1709	Pig	*Salmonella*	*enterica*	Typhimurium	Feces isolate from diseased individual
69	Human	*Listeria*	*monocytogenes*		Feces isolate from diseased individual
72	Pig	*Listeria*	*monocytogenes*		Feces isolate from diseased individual
711	Sheep	*Listeria*	*monocytogenes*		Feces isolate from diseased individual
645	Human	*Yersinia*	*enterocolitica*	O:3 6A 28	Feces isolate from diseased individual
646	Human	*Yersinia*	*enterocolitica*	O:5 27 8A 30	Feces isolate from diseased individual
647	Human	*Yersinia*	*enterocolitica*	7A 29	Feces isolate from diseased individual
N1	Pig	*Escherichia*	*coli*	F29	Feces isolate from diseased individual
N2	Pig	*Escherichia*	*coli*	F38	Feces isolate from diseased individual
N3	Pig	*Escherichia*	*coli*	F54	Feces isolate from diseased individual
144	Lab0strain	*Escherichia*	*coli*	K12	Experimental laboratory strain
N4	Human	*Staphylococcus*	*aureus*	CC1	Skin isolate from person suffering from atopic dermatitis
1077	Human	*Staphylococcus*	*aureus*		Skin isolate

## Data Availability

Data are contained within the article and Appendix A.

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
