# Peer review of "Fermented Rapeseed and Soybean Alone and in Combination with Macro Algae Inhibit Human and Pig Pathogenic Bacteria In Vitro"

_microorganisms, 2024, doi:10.3390/microorganisms12050891_

Round 1
Reviewer 1 Report
Comments and Suggestions for Authors
This research study investigated the antimicrobial potential of fermented soybean- and rapeseed-based pig feed supplements with and without added seaweed. Furthermore, it investigated the metabolite profiles based on an LC-MS analysis of the fermented products in comparison to its unfermented constituents.
The study is well-designed and the results are significant. The results revealed pronounced release of potential antimicrobial secondary metabolites such as benzoic acids when the plant material is fermented, and a significantly increased antimicrobial effect compared to the unfermented controls against several pathogenic bacteria.
I just suggest minor revisions regarding improving the English of the writing style and grammar in the manuscript. Also, I suggest re-writing and improving the conclusion section to also highlight the recommended future perspectives based on the current results obtained from this research.
Comments on the Quality of English LanguageModerate editing of English language required
Author Response
Dear Reviewer 1,
Thank you for the excellent and detailed review. Below you can find my comments to the things that you have been mentioning in your review.
Reviewer 1 comments, with reply:
I just suggest minor revisions regarding improving the English of the writing style and grammar in the manuscript. Also, I suggest re-writing and improving the conclusion section to also highlight the recommended future perspectives based on the current results obtained from this research.
I have made an extensive language and grammar review. Likewise have I rewritten the conclusion to address the future perspectives as you mentioned.
Once again thanks for you revision.
Reviewer 2 Report
Comments and Suggestions for Authors
Dear authors
Thanks for your effort and presentation.
However, some comments should be considered during revision;
- The title is so broad and dosen’t represent the aim of the study. It should be modified to become more specific. It is supposed modification to “The in-vitro antimicrobial potential of fermented pig feed supplements, based on soybean and rapeseed, with and without seaweed”.
- In the title, inhibits or inhibit?!
- Why the species of the livestock animal (pig) was not mentioned in the title?
- Why the term “in vitro” was not mentioned in the title?
- Line 16, the abbreviation “LC-MS” should be mentioned in full name. The same in line 22 “CCI”.
- The conclusion should be added to the end of the abstract.
- Why the author specifies zinc oxide for the treatment of diarrhea in pigs? Are there any other non-prohibited specific drugs as alternatives that can be used for the treatment?
- Line 56, “Xie et al.”, the year should be added.
- Ascophyllum nodosum, Saccharina latissimi, Pediococcus acidilactici, Pediococcus pentosaceus, Lactiplantibacillus plantarum, Salmonella enterica serovar Typhimurium, Listeria monocytogenes, Escherichia coli, Yersinia enterocolitica and Staphylococcus aureus should be written in an italic form.
- Line 116, A. nodosum and S. latissimi, the abbreviated pathogens should be written in full, then abbreviation. The same in line 153, L. monocytogenes and S. Typhimurium and line 292, K. oxytoca
- Lines 121-129, the bacterial isolates of human origin were not mentioned in the text. Only those of pig origin were mentioned.
- Line 154, gram (Gram).
- No figure numbers should be mentioned in the discussion.
- Only one reference has been mentioned in the discussion section! The discussion should be deep and many references should be added.
Best wishes
Comments on the Quality of English LanguageMinor English Language editing is required.
Author Response
Dear Reviewer 2,
Thank you for the excellent and detailed review. Below you can find my comments to the things that you have been mentioning in your review.
Reviewer 2 comments and my replies:
- The title is so broad and dosen’t represent the aim of the study. It should be modified to become more specific. It is supposed modification to “The in-vitroantimicrobial potential of fermented pig feed supplements, based on soybean and rapeseed, with and without seaweed”.
I have now changed the title of the manuscript, so that it clarifies that the study is an In vitro study. The new title is as follow: “Fermented rapeseed and soybean alone and in combination with macro algae inhibit human and pig pathogenic bacteria in vitro”
- In the title, inhibits or inhibit?!
This is also a good point, and I have changed the it to inhibit. To me it also sound more right without the s.
- Why the species of the livestock animal (pig) was not mentioned in the title?
It’s a good point to mention the pig in the title, and that has also been added to the title.
- Why the term “in vitro” was not mentioned in the title?
Also a good point, and the “in vitro” is now added to the title.
- Line 16, the abbreviation “LC-MS” should be mentioned in full name. The same in line 22 “CCI”.
I have now written out the abbreviation in line 16 for enhanced readability.
- The conclusion should be added to the end of the abstract.
I have revised the abstract and added the conclusive part to that.
- Why the author specifies zinc oxide for the treatment of diarrhea in pigs? Are there any other non-prohibited specific drugs as alternatives that can be used for the treatment?
We focus on zinc oxide as it has been a practice for a very long time, and now that it is prohibited, we need new strategies to prevent the development of diarrhea. Instead of using other drugs it is more favorable to improve the overall gut health in the pigs, instead of giving them drugs. Therefore we focus on fermented plant material.
- Line 56, “Xie et al.”, the year should be added.
The year of the Xie et al. paper has now been added to the text.
- Ascophyllum nodosum, Saccharina latissimi, Pediococcus acidilactici, Pediococcus pentosaceus, Lactiplantibacillus plantarum, Salmonella enterica serovar Typhimurium, Listeria monocytogenes, Escherichia coli, Yersinia enterocolitica and Staphylococcus aureus should be written in an italic form.
Together with an extensive language revision, I have also changed all the scientific names to cursive. They were originally in cursive in my draft, but I think they got changed when I copied my text into the MDPI template.
- Line 116, A. nodosum and S. latissimi, the abbreviated pathogens should be written in full, then abbreviation. The same in line 153, L. monocytogenes and S. Typhimurium and line 292, K. oxytoca
The names are now written out so that they include the full name.
- Lines 121-129, the bacterial isolates of human origin were not mentioned in the text. Only those of pig origin were mentioned.
Good point. Additional text now clarifies that we both use human and livestock pathogens in our assays.
- Line 154, gram (Gram).
This part is now adjusted in the revised manuscript.
- No figure numbers should be mentioned in the discussion.
Also a good point, and I have now adjusted the text to this.
- Only one reference has been mentioned in the discussion section! The discussion should be deep and many references should be added.
I fail to see what you mean here, as I cite multiple references in the discussion. The discussion reflects a discussion of the findings held up against findings from previous studies.
Reviewer 3 Report
Comments and Suggestions for Authors
My Comments as follow for consideration for addressing, it is very interesting but there is a little simple as whole or regular paper?
1, Abstract: some key important parameters as threholds and others, should be quantitatively noted with index here?
2, And Line 20-22: Salmonella enterica Typhi- 20 murium, Listeria monocytogenes, Yersinia enterocolitica, Staphylococcus aureus, should be typed in form of italic? And so do the similar errors in other place?
3, Line 36, a title as “Guidance on the identity, characterisation and conditions of use of feed additives” from reference 5 is advised to cite here?
4, Line 83-84, Fermentation leads to an increased release of bioavailable compounds of the plant material..., one question is this bioprocess should alos can decomposite these target compounds at the same time in deducation? any selective action of only release alone can be shown or proved?
5, Introduction, special technical reason for combiantion both ferment rapeseed and alga compound should be addressed in more details in terms of nutrition complement, advantage merging and shortage advoiding, insead of other raw materials and plants?
6, Line 132, 2.4 Plate well diffusion assay: it seems too simple than MIC methods?
7, Line 144-145 2.5 Thermal stability and resistance to proteolytic and lipolytic enzymes of antimicrobial compounds: besides these three factors, why no pH was chosen for test?
8, Safety as Hemolysis to animal as host are advised to text by suitable model cell? if possible
9, 3.1 Metabolomics analysis of the fermented plant materials assessed by liquid chroma-171 tography-mass spectrometry (LC-MS) revealed a significant increased abundance of me-172 tabolites across all analyzed structural classes in the fermented variants (Fig. 1 A-E), I did not familar with this aspect, but five kinds of basic nutrients should be presented releasly more fully such as in form of table and others?
10, Figure 2, varation error seems too high/large? why? Is this unstable from compound or fermentation or assay protocol? any reasonable explanation?
(END20240318)
Author Response
Dear Reviewer 3,
Thank you for the excellent and detailed review. Below you can find my comments to the things that you have been mentioning in your review.
Reviewer 3 comments and my replies:
1, Abstract: some key important parameters as threholds and others, should be quantitatively noted with index here?
The Abstract and the rest of the manuscript have been under an extensive revision also regarding the language that have been pointed out. I have uploaded 2 different manuscripts now. One that is showing tracking of all the changes done to the manuscript, and a more clean one that is easier to read without all the tracking.
2, And Line 20-22: Salmonella enterica Typhi- 20 murium, Listeria monocytogenes, Yersinia enterocolitica, Staphylococcus aureus, should be typed in form of italic? And so do the similar errors in other place?
During the extensive revision of the text I also changed all the formatting of the scientific names. My original draft had all the names in italic, but I think it changed the formatting when I copied it to the MDPI template. And I just didn’t notice it unfortunately. But now its corrected.
3, Line 36, a title as “Guidance on the identity, characterisation and conditions of use of feed additives” from reference 5 is advised to cite here?
This reference is now changed, as its clearly a citing mistake. Furthermore, I have gone through all of my references to make sure that they match the statements and everything is aligning now.
4, Line 83-84, Fermentation leads to an increased release of bioavailable compounds of the plant material..., one question is this bioprocess should alos can decomposite these target compounds at the same time in deducation? any selective action of only release alone can be shown or proved?
I’m not 100% sure I understand this question/comment. But if I understand it correctly, you refer to the microbial processing of compounds that are released in the fermentation. There might be a modification of the compounds in a microbial context, but this is beyond the scope of our study.
5, Introduction, special technical reason for combiantion both ferment rapeseed and alga compound should be addressed in more details in terms of nutrition complement, advantage merging and shortage advoiding, insead of other raw materials and plants?
Our research is primarily concentrated on investigating the antimicrobial properties of various compounds, rather than their nutritional profiles. While the nutritional benefits of these fermented products, that are produced by European Protein, are well-documented through previous studies, our aim is to look into their capacity to inhibit pathogenic microorganisms. This study, therefore, does not encompass the nutritional aspect, which has been adequately covered by earlier research, but instead highlights the significant antimicrobial activities exhibited by these compounds.
6, Line 132, 2.4 Plate well diffusion assay: it seems too simple than MIC methods?
MIC methods are very useful when you have single compounds or at least more compounds in a known concentration and relation, which we don’t have for the plant substrates. Therefore we did not use MIC but instead plate well diffusion assays.
7, Line 144-145 2.5 Thermal stability and resistance to proteolytic and lipolytic enzymes of antimicrobial compounds: besides these three factors, why no pH was chosen for test?
We assed proteolytic and lipolytic analysis to investigate the structural class of the compounds. pH would have been a preferable factor to test if we were looking into the physiochemical property as such. This can be done for future and more elaborate studies on the compounds that we extracted, but is deemed out of scope of this manuscript for now.
8, Safety as Hemolysis to animal as host are advised to text by suitable model cell? if possible
Cytotoxic assays could be performed, but we are not aware of the compound concentrations as such in the products, only the relative relation between the compounds. Therefore we would not be able to make direct connections between a mammal cytotoxic assays and our findings. It’s an interesting thought, and several of the compounds have already been tested extensively for their cytotoxic potential in a broad range of studies, but it is likewise considered out of scope in this study as we focus on antimicrobials.
9, 3.1 Metabolomics analysis of the fermented plant materials assessed by liquid chroma-171 tography-mass spectrometry (LC-MS) revealed a significant increased abundance of me-172 tabolites across all analyzed structural classes in the fermented variants (Fig. 1 A-E), I did not familar with this aspect, but five kinds of basic nutrients should be presented releasly more fully such as in form of table and others?
I’m not 100% sure what you mean with this comment. I have pointed out different compounds that are known from the literature to have an antimicrobial effect on the pathogens that we have tested. So that’s what we are focusing on here.
10, Figure 2, varation error seems too high/large? why? Is this unstable from compound or fermentation or assay protocol? any reasonable explanation?
This variation is hard to point out exactly, as there can be different contributions to this. Fermentation and extraction is not likely to be the greatest contributor. It can be due to the sensitivity of the bacteria when they are put into the warm agar. Even though we measure the temperature in agar, there might be a little variation, and that variation could weaken the bacteria a bit and make them more susceptible to the antimicrobial compounds they are presented to. The clear tendency of variation in antimicrobial effect when comparing fermented and unfermented is still seen though.
Round 2
Reviewer 3 Report
Comments and Suggestions for Authors
Yes, thanks for your addressing my 10 questions and could you pls answer some new questions as follows
To be continnued to my commnets of 1st turn
11, Table 1, from soybean source, three fermented products, their key quality index as antinutrient contents as glycinin, beta-conglycinin, and trysin inibitor, stachyose and raffinose, before and after fermentation, should be supplemented here, it is basic necessary for reader even thogh they can be probably seen from references, yes, this work related to commercial goals, but basic indigrents disclose and list is required by academic paper and so do your product introduction as EP200 and EP 2299, just copy citation here;
12, Table 1, yes, key antinutrient compoents in fermented and unfermented rapeseed as glucosinolate and tannin, should be noted here;
13, And the limitation standard of above those anitinutrients in pig additive and combination feed in EU or Denmark regulation?
14, Table 2, strain 69,72,711, 645,646,647, you should note their sources like others ;
15, Figure 2, Sid-by-side or side-by side? pls confirm it?
16, 3.1 whole section and Line 468-476, as to Figure 1 as a key Figure, this is very informative data, most readers and reviewer probably want to get information analysis on critical point or transition point and quantification description as highest, lowest, span, scope and others from key indexes instead of qualitative word as high or low and increase or decrease? And more explannation and description is highly hoped; Yes, you can certainly, I am sure.
17, Improvement on protein prodcuts quality from three sources, and special contribution from three lacto bacteria and their fermentation should be highlihted more, their biological and fernmetation relationship should be enhanced more.
18, If possible, any standard product as control like HP2000 or other similars should be refered and compared from your trials or references and other publications?
(end at 19 Apr 2024)
Author Response
11, Table 1, from soybean source, three fermented products, their key quality index as antinutrient contents as glycinin, beta-conglycinin, and trysin inibitor, stachyose and raffinose, before and after fermentation, should be supplemented here, it is basic necessary for reader even thogh they can be probably seen from references, yes, this work related to commercial goals, but basic indigrents disclose and list is required by academic paper and so do your product introduction as EP200 and EP 2299, just copy citation here:
Thank you for your valuable feedback. I understand your point regarding the inclusion of detailed data on antinutrient contents such as glycinin, beta-conglycinin, and trypsin inhibitor, as well as stachyose and raffinose, in Table 1. While these data are indeed a valid background for a comprehensive understanding of the fermentation process, our current dataset is incomplete in this regard and does not cover all the products tested in our study comprehensively. Consequently, we will not be delving deeper into these parameters within the scope of this specific research, as our primary focus remains on evaluating the antimicrobial effects. However, we acknowledge the importance of this data and will include the available information as supplementary data to support the study's findings and provide additional context where possible.
12, Table 1, yes, key antinutrient compoents in fermented and unfermented rapeseed as glucosinolate and tannin, should be noted here.
Avaliable data on this matter is being added as supplementing data, as described in the above comment.
13, And the limitation standard of above those anitinutrients in pig additive and combination feed in EU or Denmark regulation?
This research is specifically focused on investigating the antimicrobial effects, and as such, we have chosen to concentrate our efforts primarily on this area. Therefore, we will not be delving into the antinutrients in detail within the context of this study. Our objective is to maintain a narrow focus on antimicrobials to ensure clarity and depth in our findings. While questions about antinutrients are undoubtedly important and merit thorough investigation, they would be more appropriately addressed in a study specifically tailored to explore those particular aspects. By keeping our research focused, we aim to provide more precise and impactful insights into the antimicrobial properties we are examining.
14, Table 2, strain 69,72,711, 645,646,647, you should note their sources like others ;
This information has now been added to the table
15, Figure 2, Sid-by-side or side-by side? pls confirm it?
This has now been corrected to Side-by-side comparison
16, 3.1 whole section and Line 468-476, as to Figure 1 as a key Figure, this is very informative data, most readers and reviewer probably want to get information analysis on critical point or transition point and quantification description as highest, lowest, span, scope and others from key indexes instead of qualitative word as high or low and increase or decrease? And more explannation and description is highly hoped; Yes, you can certainly, I am sure.
Im not completely sure I understand this comment. The process involves analyzing each compound to evaluate its relative abundance across different sample types. This evaluation is conducted through trend analysis, where we systematically observe the rises and falls of specific compounds within these samples. This approach helps us understand how fluctuations, likely caused by the fermentation process, affect the presence of these compounds. To simplify the analysis, we categorize these compounds into different metabolite classes. This categorization provides a clearer view, emphasizing how fermentation influences the abundance of various metabolites in the final products.
17, Improvement on protein prodcuts quality from three sources, and special contribution from three lacto bacteria and their fermentation should be highlihted more, their biological and fernmetation relationship should be enhanced more.
Thank you for your insightful suggestion regarding the enhancement of our discussion on protein product quality improvements from three sources and the specific contributions of three lactobacilli in fermentation. Your point about deepening the exploration of the biological and fermentation relationships is well-taken. However, our study's current scope is primarily focused on other specific aspects of fermentation, which has limited our ability to extend our analysis to these areas in depth. Nonetheless, we recognize the importance of this aspect and suggest it as a promising direction for future research. We appreciate your understanding and agree that further investigation into these relationships could provide valuable insights into protein product enhancement through fermentation.
18, If possible, any standard product as control like HP2000 or other similars should be refered and compared from your trials or references and other publications?
You raise a good point. However, our research specifically aims to explore the differences in antimicrobial effects between an unfermented and a fermented version of the same substrate. This focus is why we have chosen not to include commercially available products in our study, as they would not directly contribute to our investigation of these specific changes. Additionally, regarding HP2000, which you mentioned, I have been unable to find any relevant information on this product. Therefore, comparing it or other similar products would not be pertinent to our focused analysis on the specified products in this particular study.
kind regards